# Improving the Performance Properties of Eutectoid Steel Products by a Complex Effect

**DOI:** 10.3390/ma15238552

**Published:** 2022-11-30

**Authors:** Vitaliy Vlasovets, Tatiana Vlasenko, Stepan Kovalyshyn, Taras Shchur, Oleksandra Bilovod, Lyudmila Shulga, Mariusz Łapka, Milan Koszel, Stanisław Parafiniuk, Leszek Rydzak

**Affiliations:** 1Department of Mechanical Engineering, Lviv National Environmental University, V. Valyki Street 1, 80381 Dubliany, Ukraine; 2Department of Management, Business and Administration, State Biotechnology University, Alchevsky St. 44, 61002 Kharkiv, Ukraine; 3Department of Cars and Tractors, Lviv National Environmental University, V. Valyki Street 1, 80381 Dubliany, Ukraine; 4Department of Industry Engineering, Poltava State Agrarian University, St. Skovoroda 1/3, 36003 Poltava, Ukraine; 5Faculty of Production and Power Engineering, University of Agriculture in Krakow, 30-149 Krakow, Poland; 6Department of Machinery Exploitation and Management of Production Processes, University of Life Sciences in Lublin, Głęboka 28, 20-612 Lublin, Poland; 7Department of Biological Bases of Food and Feed Technologies, University of Life Sciences in Lublin, Głęboka 28, 20-612 Lublin, Poland

**Keywords:** eutectoid steel, modification, microstructure, watershed segmentation, 2D image

## Abstract

This study focuses on the assessment of possible hypereutectoid steel carbide mesh crushing. It is used for tools production, including forming rolls of various diameters, with modification and cyclic heat treatment methods. For steel containing 1.79–1.83% C, we studied the effect of 0.35–1.15% Si on the possible crushing of the cementite mesh within crystallization by introducing modifiers Ti, V, N, as well as simultaneously modifying V with N and Ti with N. The obtained castings of Ø200 mm, 400 mm high were cut into discs, from which we made samples for tests on wear, determining mechanical properties, thermal resistance, and susceptibility to brittle fracture. The assessment was performed in the as-cast and after double and triple normalizing and annealing with drawback. With additional fans blowing, we changed the cooling rate from 25 °C/h to 100–150 °C/h. We performed the microstructure analyses using traditional metallographic, micro-X-ray spectral analyses, and also used the segmentation process based on 2D image markers. It was found that the as-cast modifying additives infusion is insufficient for carbide mesh crushing. It can be made by multi-stage normalizing with accelerated cool-down for products up to 600 mm in diameter to cycle temperatures above the steel transfer from a plastic to elastic state (above 450 °C).

## 1. Introduction

The growing demand for rolled metal products quality is continuously connected to the fail-safe operation of forming rolls [1]. The analysis of the related publications on improving mechanical properties of the products’ working layer shows that improvements are carried out in the following main directions: optimization of the used materials chemical composition and development of new materials [2,3]; effective heat treatment methods development [4,5]; and development of the reliable methods for the massive product’s working layer quality control without any destruction [6,7,8].

The existing traditional methods of non-destructive testing (eddy-current, radio-wave, thermal, optical, radiation, and acoustic) are mainly directed to search for and find a concrete defect [9,10]. Moreover, the depth and extent are also determined [11].

However, the above-mentioned methods fix only those defects that have already been identified [12]. More effective assessment consists of assessment of the structural state within the production using a non-destructive magnetic method. Such assessment allows for choosing the optimization for the required properties set. At the operating stage, this method can assess the stability of the structural state and properties [13]. When necessary, it also can perform a regulatory action based on the available data to restore the functioning state (mechanical processing, coating). The most proper operational assessment of the magnetic methods (magnetization, residual induction, magnetic permeability, Barkhausen effect) is the coercimetry method with Hc determination [14].

Currently, due to the significant influence of changing factors, there is no general theory of magnetic structuroscopy. Therefore, in each specific case, it is necessary to find a relationship between magnetic and other properties of the respective material [15]. Even in such cases when this relationship is known from literary and other sources [16], it is still necessary to examine and establish conditions for its existence by implementing control under specific conditions of the production.

We observed a great number of studies concerning new resistant materials [17]. They are developments of widely used materials optimized through their additional alloying and modifying, as well as of new steels [18].

The heat treatment methods are directed to strengthen the product’s working layer [19]. The heat treatment modes of high carbon steels are divided into single-stage and multi-stage modes, depending on their purpose [20].

The single-stage modes are used in the production of rolls for finishing and pre-finishing stands. They include surface temper hardening, artificial aging, and annealing of the I type. The rolls of roughing stands and cogging mills need high strength requirements; therefore, hypereutectoid steels can be used only after multi-stage heat treatment, which ensures maximum carbide phase crushing, grain breaking, and pearlite spheroidizing [21].

The high-carbon steels are heat treated to relieve stress and increase strength and ductility, or hardness, of the material. To relieve stress, we use low-temperature annealing at 500–600 °C or double tempering after high-temperature heat treatment.

An increase in strength and ductility can be achieved by means of implementing [22] cyclic treatments and treatments in the magnetic transformation temperature range. These methods lead to the formation of a block structure. Such methods are recommended for tool and carbon steels. There is no available information on the possibility of their use for hypereutectoid steels when the carbide mesh crushing is the first matter. Only then it is possible to choose the perlite matrix spheroidizing and substructural strengthening methods.

The heat treatment [23,24] eliminating solid cementite mesh and, at the same time, preserving separate spheroidized inclusions, provides the following values: σ_B_ = 1176 MPa, δ = 2–3.0%, roll hardness–380–390 HB. The absence of cementite large inclusions, as well as obtaining the structure of troostite and sorbite with small carbides uniform distributing in the matrix, makes it possible to obtain the hypereutectoid steel of the following values: σ_B_ = 1274 MPa, δ = 2–3.0%, roll hardness up to 400 HB.

The multi-stage heat treatment of the steel cast rolls is very difficult due to the I and II types of stress occurrence, which significantly rises with the increase in a roll diameter and rate of cooling [25].

The performed materials review shows that there are no available data on effective heating and cooling rates, optimal ratios of structural components, multi-stage processing, and tempering, which ensure achievement of the required operational properties set. There is also no available information about intensity and significance of the II type stress occurrence. 

The heat treatment mode efficiency is determined not only by the operation requirements basis [26], but also by its manufacturability and durability, as well as cementite mesh crushing intensity. Known heat treatment modes do not always allow high carbon steels to be used for pre-finishing and finishing stands, as they do not meet the wear resistance requirements. At the same time, such materials’ application for break-down stands is also limited, especially when it is necessary to achieve good metal pickup during rolling.

The objective of this research is to improve the performance properties of eutectoid steels by their modification and heat treatment, which determine the carbide phase formation during casting, the carbide mesh crushing to enhance the products durability.

## 2. Materials and Methods

For steel with 1.8% C, we planned to establish a silicon influence on the tendency to form as-cast graphite and the cementite mesh crushability during crystallization by modifiers Ti, V, and N infusion. For this purpose, we performed two melts. Each of them was poured into three ladles, one cast at a time. Within the first melt, the first ladle metal was original, the second and third were V- and Ti-modified, respectively. Within the second melt, the first ladle metal was also original, and the second and third were V-, N- and Ti-, N-modified, respectively. Table 1 shows the chemical composition of the casts with a diameter of 200 mm and a height of 400 mm.

Alloys composition was determined by chemical (C, O, N) and spectral (Mn, Cr, Si, P, Ni, Cu, Al, Ti, Mo) methods, as well as refined by micro-X-ray spectral analysis.

Assessment of the composition of the crystallizing phases and the dispersed phases distribution was carried out on scanning electron microscopes microanalysts (SEM, JSM–6390LV, JEOL, Tokyo, Japan) and (SEM, JSM–820, JEOL, Tokyo, Japan) with the Link AN10/85S “Link Analytical” X-ray microanalysis system. At the same time, the effective spot size was 2–3 microns. To assess the distribution of chemical elements, we performed a spectral analysis (cauterization spot diameter up to 2 mm) on the XRF Analyzer (XRF Analyzer, X–Met 3000TXR, Oxford Instruments, Abingdon, UK) and multi-channel optical emission spectrometer (OES, SPECTROLAB F8, Kleve, Germany).

The carbide phase analysis was performed on samples Ø10 mm, l = 70 mm by the carbides weight electrolytic isolation method. The electrolyte composition is 730 cm^3^ of water and 270 cm^3^ of hydrochloric acid, and the specific weight is 1.19 and 25 g of citric acid. The carbides were isolated at room temperature held for 2–2.5 h, current density 0.02–0.25 A/cm^2^.

We determined the elements’ content in the filters and sediments, obtained after the above treatments, using known chemical methods.

The metal temperature (induction melting furnace, ICMEF-0.03/0.05 ТrМ, Termolit, Kyiv, Ukraine) in the furnace and ladle was 1495–1510 °C (thermocouple Type R) and 1490 °C respectively. The metal was cast at 1450–1465 °C. The casts were cut into discs, from which we made the samples in order to test their wear, mechanical properties, thermal resistance, and tendency to brittle fracture. The tests were performed in the as-cast state and after the various types of heat treatment (double normalizing 950 ± 10 °C, 850 ± 10 °C; triple normalizing 950 ± 10 °C, 900 ± 10 °C, 850 ± 10 °C; and annealing at 850 ± 10 °C with tempering).

The materials’ wear was assessed by testing disks of a 6 mm diameter with friction under the load of 9.8 MPa (Table 2). Due to the small contact area, it was possible to provide pressure in the surface layer up to 350–400 MPa (pilot installation, Kharkiv, Ukraine). At the same time, the temperature of the disk (induction heating) imitating the rolled metal was 800 ± 20 °C (pyrometer, WT900, Wintact, Shenzhen, China), and 40–45 °C of the test material.

After the heat treatments, we determined the impact toughness by the Charpy impact tests (impact testing machine, ТСКМ-300, Kiev, Ukraine) at room temperature. There were prepared tensile samples with the dimensions 10 mm × 10 mm × 55 mm according to the U-notch impact specimens for the Charpy impact tests at room temperature.

The thermal endurance tests were carried out on pinched samples, which provided the specified alternating stresses. They were controlled within standard mechanical tests. We took the number of cycles before the samples’ destruction as thermal endurance, recording them automatically. The cycle time was 35 s (heating 30 s, cooling 5 s). The recording of thermal stresses arising in a sample during cyclic processing was performed by a recording potentiometer.

Assessment of the microstructure was carried out on micro–grinds and directly on the products themselves, after surface preparing, on a metallographic microscope (OM, MIM-8M, Ukraine) and a portable TCM (OM, TCM, Ukraine) at magnification ×100, ×200, ×500, and ×1000. The analyzed surfaces were etched with a 4% solvent of nitric acid (HNO_3_) in ethyl alcohol.

According to the AUSS E384-17 Standard Test Method for Microindentation Hardness of Materials, the distance from the indenter imprint to the measured structure boundary within microhardness assessment should be at least three indentation diagonals. As it is shown by statistical and special studies, this condition is not necessary when the microhardness exceeds HV_50gf_ = 600. With this high hardness, the grain boundary does not affect the measurement results. This made it possible to assess this characteristic in high-carbon alloys, in which the carbide phase is a needle or a thin mesh of the secondary cementite.

The structurally sensitive method of coercive force (H_c_, A/cm) measuring was used as a method of the product non-destructive quality control. The method of magnetic structurescope-coercitimeter measurements is given in the corporate standard 29.32.4-37-532 developed by us. The measurements were carried out using portable magnetic structuroscope–coercitimeter (MSC, КRМ-C, Kharkiv, Ukraine). The device–КРМ-Ц is designed to measure the coercive force of the ferromagnetic material’s local area. The device can measure a range of coercive force–1.0–60.0 A/cm and has a measurement error of coercive force on control samples of not more than 2.5%.

The device operation principle is based on the H_c_, the measured current calculation of the residual magnetic density offset in a closed magnetic circuit. The circuit is created by a magnetic converter system, the poles of which are closed by the controlled sample.

The measurement cycle includes magnetic preparation (duration 2 s), residual magnetization compensation (2 s), H_c_ calculation, and result indication.

During the analysis preparation, the studied product area between the pole tips of the magnetic converter system was periodically magnetized for saturation with current pulses of amplitude to at least 3.0 A. The residual magnetization field was then automatically compensated. The value was calculated according to the current magnitude required to create a compensating magnetic field. After that, the digital display was turned on.

## 3. Results and Discussion

### Heat Treatment Modes and Hypereutectoid Steel Properties 

In hypereutectoid steel products, combined methods are used to ensure the carbide mesh thinning and crushing, as well as to increase the core strength. In the casting process, this is partially provided by the modifying additive of vanadium up to 0.45% or titanium. Cementite mesh crushing during the heat treatment was carried out by the multi-stage normalizing. In this case, each heating stage had a temperature lowering, and cooling was carried out at an accelerated rate to avoid previously fragmented carbides coagulation along the grain body and mesh. The single-stage processing does not assist the mesh crushing, and three-stage processing regulates the hardness and the crushing degree depending on the accepted heating temperatures combination.

The double normalizing with accelerated cooling ensures the sufficient cementite mesh crushing in the surface layer. For the required structure formation in the core and necks, it is advisable to carry out modification with vanadium or titanium. A thinner mesh is achieved by limiting silicon to 1.0% during casting of the low-alloy high-eutectoid steel. 

An increase in the Si concentration in the steel with 1.8% C was found to reduce the wear resistance (see Table 2). A 43% decrease in wear was characteristic of the third cast, which was Ti-modified. We identified the mechanical characteristics, according to which an increase in H_c_ by 10 A/cm and in hardness by 10 HB expands an increase in the weight loss difference by 2.8% and 0.4% respectively.

During the casts’ heat treatment, we studied samples, the parameters of which corresponded to the working layer of real rolls (Figure 1) using triple (from 950 ± 10 °C, 900 ± 10 °C, 850 ± 10 °C), double (from 950 ± 10 °С, 850 ± 10 °С) normalizing, and annealing (from 850 ± 10 °С) with tempering (from 600 ± 10 °С). The total treatment time was 210 h, 140 h, and 75 h respectively. Double normalizing, tempering, and annealing were performed with the same heating and cooling parameters as for triple normalizing (Figure 1). To assess the carbide phase degree of crushing and coagulation, we studied the heat treatment parameters’ influence on mechanical properties and Н_с_ (Table 3 and Table 4).

The steel with a minimum concentration of silicon (0.36–0.38%) and 1.8% C in the as-cast state tended to have increased strength characteristics (σ_B_, σ_0.2_) and hardness with the infusion of additional modifiers. A slight increase in thermal resistance is a characteristic of the V-modified steel. Chemical composition and heat treatment affect not only the metal matrix, but also the carbide phase. Thus, when 0.45%V is infused, the carbide phase microhardness level increases on average from HV_50gf_ = 840 to HV_50gf_ = 998 (by 19%). The unmodified steel heat treatment (annealing 850 °C) increases the cementite microhardness by the same proportion. With accelerated cooling from 1050 °C, it reaches HV_50gf_ = 1040. The double normalizing with tempering reduces microhardness to HV_50gf_ = 930. At the same time, the maximum spread of values is typical for steels after tempering and is 13–14%.

The infusion of 0.03% Ti leads to its double decrease (with an increase in σ_B_ and σ_0.2_ by 10% and 7%, respectively). This is associated with the increased metal tendency to cracking. The Ti infusion cancels the graphitization process, promotes the dispersed carbides formation, and increases the H_c_ magnetic parameter level as more additives are present in steel. 

The heat treatment significantly changes the steel structure. Annealing leads to changes in matrix structure as well as in properties. Instead of appearing as lamellar in the as-cast state, they are formed in the granular perlite areas, which lowers the H_c_ level by 14–30%. Secondary carbides are shown to be against the pearlite matrix background.

Annealing also partially changes the primary carbides shape, i.e., isolated, more rounded particles are formed in place of the secondary cementite needles. Nevertheless, the partially carbide mesh was presented. This is especially typical for annealed unmodified metal.

Double normalizing, as well as annealing, leads to the granular perlite structure formation in the matrix, as well as to the secondary carbides individual inclusions crushing.

The heat-treated steels matrix with the sorbitol structure of uniformly distributed fine spheroidized carbides is characterized by microhardness HV_50gf_ = 270–280. A decrease in hardness after the heat treatment is mainly determined by the cementite mesh crushing.

The modified steel with 1.08–1.15% Si shows a reduction in all strength characteristics and thermal resistance. 

Given that the studied steel rolls are used only after the heat treatment, we assessed the strength and plastic characteristics, as well as thermal resistance after studying its various types.

We determined the relationships for the as-cast state, double and triple normalizing, and annealing, which makes it possible to assess the basic properties level in terms of Н_с_ (Table 4).

The gray-level micro-X-ray spectral analysis images of the hypereutectoid steel have several components, namely: a carbide phase of the cementite type of variable stoichiometric composition; coarse primary carbides, which form a carbide mesh; an alloy matrix with a perlite component; a compounded carbide phase; and an inclusion of graphite (Figure 2a). The attenuation coefficient of the X-ray (grayscale intensities) has the maximum value (darker grayscale intensities) in comparison with the steel matrix. In the carbide phase, the attenuation coefficient is lower than in the matrix, and decreases with the increase of the carbon content (lighter grayscale intensities). The threshold value was set for the local minimum to alleviate catchment basins phases of creation.

The pair comparison of free terms in the equations (Table 4) shows that the coarse carbide mesh presence in the as-cast state, which, as proved by metallographic studies, cannot be fragmented by Ti-, V-, and N-modification, determines the level of the initial state properties. At the same time, it increases the hardening intensity after annealing. This indicates a possible more efficient use of such modifiers to control the properties after high-temperature annealing. However, an increase in strength properties and hardness lead to a decrease in the metal thermal resistance. The infusion of 0.03% Ti leads to its double decrease (with an increase in σ_B_ and σ_−0.2_ by 10% and 7%, respectively). The double (950 ± 10 °C, 850 ± 10 °C) and triple normalizing (950 ± 10 °C, 900 ± 10 °C, 850 ± 10 °C) are directed to the carbide mesh crushing. Such treatment provides the less equalized structures’ formation with a higher basic (free term in the equations, see Table 4) level of properties. 

A random walker algorithm completes the segmentation [27] of the gray-level micro-X-ray spectral analysis images of the hypereutectoid steel from a marker's set identifying five components for the steel. An anisotropic diffusion equation is explained with the marker’s tracers begun at their position. The local diffusivity number is greater if neighboring pixels have similar values. The sign of each unknown pixel is coupled to the sign of the known marker [28]. This has the highest probability to be achieved first within this diffusion process [29].

The images were denoised using the non-local means filter [30]. The non-local means algorithm replaces the pixel value by a selective average of other pixels values. Small patches centered on the other pixels are compared to the patch centered on the pixel of interest, and the average is performed only for pixels that have patches close to the current patch [31]. As a result, this algorithm can properly restore textures from the level micro-X-ray spectral analysis images.

We established an experimental relationship for estimating the yield stress after annealing in Н_с_ in the range of 5.2–7.4 A/cm; 460–550 MPa:σ_0.2_ = 218 + 37.6 H_c_(1)

The structural-phase states after annealing and casting are the closest ones to equilibrium. At the same time, double and triple normalizing results in the formation of the less equalized structures and a higher stress level during casts cooling. Therefore, it is advisable to compare the relationships for evaluating the paired properties.

The normalizing does not lead to the carbide mesh intense crushing, which indicates the change intensity increase in the ultimate strength (Table 4) compared to casting and annealing (by a factor of 2.5–11) in a given range of changes in Н_с_ (7.5–10.3 A/cm). The physical meaning of free terms, reflecting the property’s base level, is missed with such change intensity in the property. 

The heat treatment was found to lead to a decrease in the average hardness (see Table 4): annealing by 22%, triple normalizing by 17%, and double by 12%. At the same time, the steel strength characteristics are increased regardless of the silicon concentration.

The double normalizing leads to an increase in σ_w_ in the original and V-modified steel in comparison with the as-cast and Ti-modified state by 34% and 51%, respectively (see Table 4). With the triple normalizing, σ_B_ increases for the original, V-, and Ti-modified steel by 12%, 18%, and 34%, respectively. A slight tendency to an increase in σ_B_ in comparison with the as-cast state is also characterized for the annealed steel at 850 °C. In this case, the effect is much more noticeable for Ti-modified steel, where σ_B_ increases by 22%.

The heat treatment also leads to an increase in σ_0.2_. The triple normalizing makes this characteristic in the original steel increase by 43%, in Ti-modified by 54%, and, to a lesser extent, in V-modified by 20%. With the double normalizing, σ_0.2_ of the original steel, regardless of the Si content, increases on average by 36%, V-modified by 23%, and Ti-modified by 63%. Annealing leads to an increase in σ_0.2_ in all studied casts by 23–33%.

To study the dissolution kinetics of the carbide mesh, we examined the samples after each normalizing step (Figure 3). It was found that single heating to 850 ± 10 or 950 ± 10 °C leads to partial dissolution of the secondary cementite needles.

Regardless of the treatment type, modification, especially with Ti, reduces the steel tendency to brittle fracture (Table 5).

Comparison of the properties of the heat-treated steel demonstrated zero special advantages of the triple normalizing over the double normalizing, both in terms of the cementite mesh crushing, as well as mechanical properties, hardness, and thermal resistance.

Both normalizing types significantly increase the thermal stability basic level and make it more sensitive to modification, especially with N. The change in properties upon N-modification, in which atoms are interstitial, provides a solid solution to significant hardening in comparison with the interstitial atoms. 

As comprehensive studies show, regardless of the heat treatment type, the steel with 1.8% C does not undergo complete crushing of the coarse carbide mesh. An increased silicon concentration of 1.0–1.5% does not provide cementite destabilization when heated up to 950 ± 10°C. Graphite also does not precipitate.

We assessed the elements’ distribution by the hypereutectoid steel (1.83% C; 1.08% Si; 0.69% Mn; 0.84% Ni; 0.74% Cr; and 0.53% Mo) structural components in the as-cast state and after the double normalizing (Figure 4 and Figure 5).

The as-cast cementite carbide, being a carbide mesh part, contains: 3.30% Cr; 1.02% Mn; 0.19% Ni; 0.83% Mo, and the remainder is iron (see Figure 2a). The zone directly adjacent to the carbide is depleted of Cr, but contains 1.05% Si, 0.13% P; 0.24% Cr; 0.27% Mn; and 0.72% Ni. Compared to the depleted area, the pearlite areas contain an increased amount of silicon, manganese, and nickel: 1.17% Si; 0.59% Cr; 0.83% Mn; and 0.80% Ni. In the steel, we also found the as-cast manganese sulfides, often located next to the cementite inclusions, and containing 26.65% S; 0.14% Cr; and 44.09% Mn. Separate carbides, compared to those in the carbide mesh, contain less chromium and manganese: 2.69% Cr; 0.85% Mn; 0.19% Ni; and 0.66% Mo.

The double normalizing indicates insignificant redistribution of components (see Figure 3). The concentration of Cr in the carbide phase increases by 20%. The carbide mesh contains 3.96% Cr and 1.0% Mn. This carbide phase contains no molybdenum and nickel.

Compared to the as-cast, the areas directly adjacent to the carbide phase also have lower concentrations of Si, Cr, Mn, and Ni. They contain 0.63% Si; 0.56% Cr; 0.65% Mn; and 0.82% Ni. The pearlite area is also depleted of silicon and manganese, but contains high concentrations of nickel and chromium: 1.10% Si; 0.71% Cr; 0.64% Mn; and 1.01% Ni. Compared to the as-cast, coagulated cementite inclusions contain more chromium and manganese and almost no nickel: 3.64% Cr; 0.90% Mn; and 1.0% Mo.

## 4. Conclusions

The structure studies after the double normalizing at 950 and 850 °C also showed zero mesh crushing. In a number of cases, it even became coarser. This fact may indicate that, within the heat treatment, the carbide mesh coarsens due to the insufficient cooling rate under intense diffusion of carbon from dissolved secondary cementite. To prove this, we performed water-quenching of the samples heated to 950 ± 10 °C. Such treatment showed no noticeable dissolution of the mesh, but significant dissolution of the secondary cementite needles. This confirms the provided statement.For more complete mesh dissolving and reducing its possible subsequent separation, we performed steel normalizing at 950 ± 10 °С (fan cooling, υ_cool_ = 100–150 °С/h) and double normalizing at 950 ± 10 °С and 900 ± 10 °С (air cooling, υ_cool_ = 25 °С/h). In the first case, the mesh was completely fragmented, while in the second case, it was much less fragmented. This confirms the need for accelerated cooling after each stage of normalizing. However, the accelerated cooling use is possible only at the heat treatment of rolls with a max diameter of 600 mm.It was found that the as-cast modifying additives for the steel with 1.8% C and 0.35–1.15% Si do not provide the carbide mesh crushing. It can be made by the multi-stage normalizing with accelerated cooling (υ_cool_ = 100–150 °C/h) up to the cycle temperatures above the steel transfer from a plastic to an elastic state (above 450 °C).

## Figures and Tables

**Figure 1 materials-15-08552-f001:**
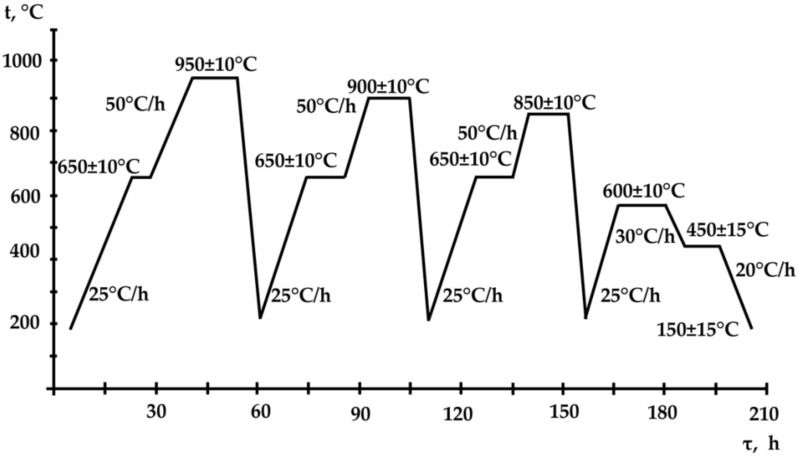
Mode and parameters of the triple normalizing, annealing of samples Ø200 mm.

**Figure 2 materials-15-08552-f002:**
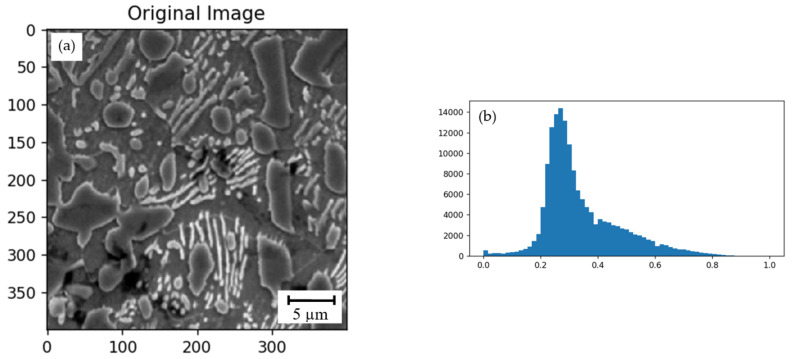
Illustrations of the marker-based watershed segmentation workflow on a 2D image of the hypereutectoid steel microstructure with 1.8% C after annealing (contains all structural components). (**а**)—the gray-level structure of the steel after annealing (etching with 4% HNO_3_ solvent); (**b**)—the histogram of gray values image after non-local means denoising and equalizing for preserving textures; (**c**)—markers image after exposure (labeling 5 components of phases); (**d**)—the segmented image with regions resulting from the marker-based watershed segmentation (the color corresponds to the segment; black–the structurally free graphite zones; blue-сement plates in perlite and matrix areas are close in content to the averaged value by perlite component; red-carbide phase with non-equilibrium stechiometric composition (depleted carbon) and alloy steel matrix; green-depleted by the steel matrix components content; azure-carbide phase with a steichiometric composition close to the equilibrium state).

**Figure 3 materials-15-08552-f003:**
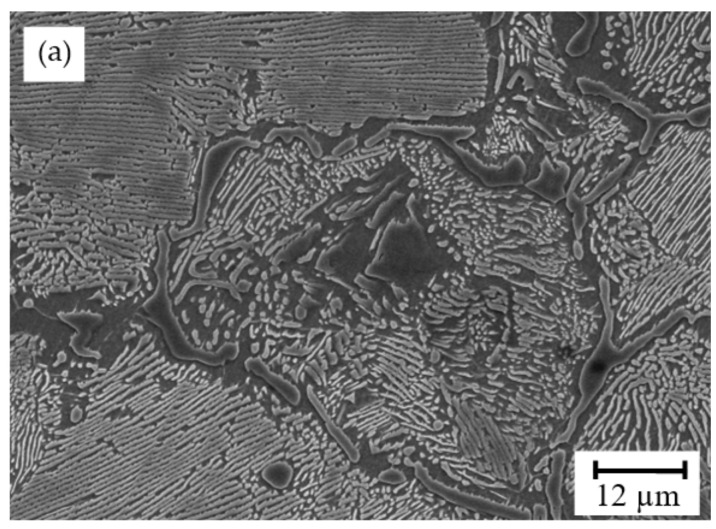
Microstructure of the hypereutectoid steel with 1.8% C. (**а**)—the as-cast state, the carbide mesh areas; (**b**)—after double normalizing; (**c**)—after triple normalizing; (**d**)—after annealing; (**e**)—manganese sulfides. Etching with 4% HNO_3_ solvent.

**Figure 4 materials-15-08552-f004:**
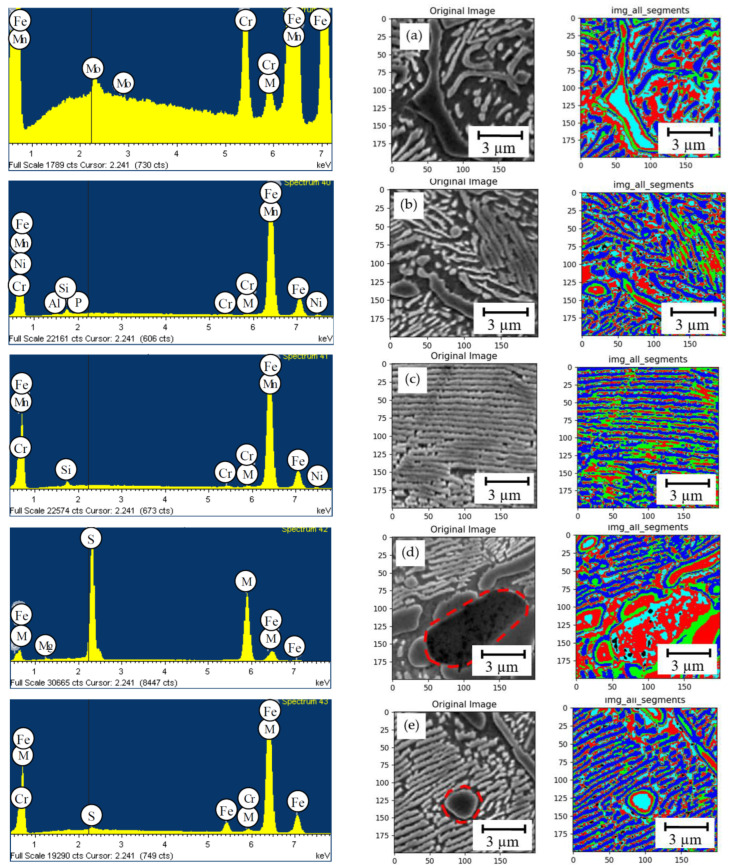
The components’ distribution according to the structural components of the hypereutectoid steel with 1.8% C at the as-cast state. (**а**)—cementite, which is a part of the carbide mesh; (**b**)—depleted area next to the cementite mesh; (**c**)—perlite; (**d**)—manganese sulfide next to the graphite inclusion; (**e**)—isolated cementite inclusion.

**Figure 5 materials-15-08552-f005:**
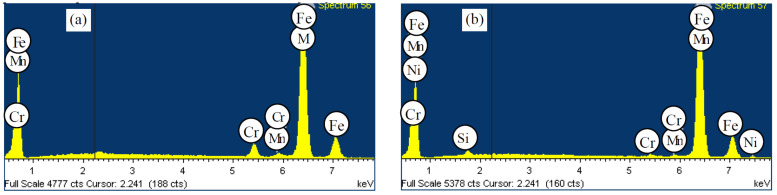
The components’ distribution according to the structural components of the hypereutectoid steel with 1.8% C after double normalizing. (**а**)—cementite, which is a part of the carbide mesh; (**b**)—depleted area next to the cementite mesh; (**c**)—perlite; (**d**)—coagulated carbide of Me_3_C type.

**Table 1 materials-15-08552-t001:** Chemical composition of the hypereutectoid steel (wt.%).

Melt No.	Cast No.	C	Si	Mn	Ni	Cr	Mo	Ti	V	N
1	1	1.83	0.36	0.49	0.85	0.69	0.52	–	–	–
	2	1.81	0.38	0.48	0.83	0.79	0.54	–	0.45	–
	3	1.79	0.37	0.46	0.85	0.73	0.54	0.03	–	–
2	4	1.83	1.08	0.69	0.84	0.74	0.53	–	–	–
	5	1.82	1.15	0.68	0.85	0.74	0.53	–	0.42	0.01
	6	1.80	1.15	0.64	0.85	0.78	0.54	0.08	–	0.01

Note: The concentration of S and P is 0.02–0.03%.

**Table 2 materials-15-08552-t002:** Wear tests of the cast samples.

Cast No.	Disc Weight, kg		Difference in Weight	
	Before Tests, kg	After Tests, kg	kg	%
1	4.138937	4.130160	0.008770	2.1
2	4.450992	4.442040	0.008952	2.0
3	4.618915	4.613520	0.005395	1.2
4	4.437749	4.425300	0.012449	2.8
5	4.44020	4.428500	0.01170	2.6
6	4.230812	4.218610	0.012202	2.8

**Table 3 materials-15-08552-t003:** Mechanical, physical properties and coercive force (Н_с_) of the hypereutectoid steel after cast and heat treatment.

Cast No.	σ_B_, MPa	σ_0.2_, MPa	KCU J/cm^2^	Hardness HB	Coercive Force А/cm	Thermal Resistance (N_cycle_)
As-cast state						
1	430–465	400–460	50–60	325	6.5–6.8	252–1482
2	465	430–460	60	329–377	7.3–7.7	168–1688
3	475–540	450–480	60–70	333–337	7.9–8.3	44–811
4	475–570	460–550	50–60	333–341	6.8–7.3	235–3042
5	485–520	440–475	60	333–341	8.5–9.0	882–1894
6	450–510	460–580	50–60	329	8.5–8.9	346
Post double normalizing						
1	456–585	560–660	70	302–321	7.9–8.2	468–9852
2	515–690	520–620	60–70	309–329	8.5–8.9	3230–14745
3	690–715	840–870	70–90	295–302	9.4–9.8	911
4	635–680	610–720	60–70	313–329	8.5–8.8	3180
5	520–640	540–580	70	321–345	9.1–9.4	391–2928
6	740–785	610–630	70–120	302–317	9.9–10.3	881–2012
Post triple normalizing						
1	510–700	610–750	60–70	260–275	7.5–7.8	1361–2550
2	482–600	490–560	60–70	269–306	8.0–8.2	452–2169
3	600–650	760–820	90–11	266–288	8.9	1762
4	496–660	630	70–80	298–317	7.8–8.0	450–2640
5	535–650	550–580	60–70	298–329	8.7–8.9	72–213
6	680–705	600–720	60–90	295–306	9.1–9.2	1249
Annealed						
1	440–530	550–580	60–70	234–255	5.2–5.6	2950
2	345–475	560–590	70–80	239–249	6.0–6.3	1140
3	510–540	560–580	70–150	236–252	6.8–7.2	2373
4	540–560	570–630	70	263–275	5.5–5.9	1010–3062
5	500–520	510–570	70	260–282	6.2–6.5	260–282
6	530–620	600–780	70–80	260–282	7.1–7.4	260–282

Note: σ_B_–ultimate tensile strength (MPa); σ_0.2_–ultimate bending strength (MPa); HB–Brinell hardness; N_cycle_-thermal resistance was defined as a number of cycles to failure (N_cycle_); KCU–U-notched impact toughness (J/cm^2^).

**Table 4 materials-15-08552-t004:** Assessment of the hypereutectoid steel main mechanical properties by H_c_.

Treatment Mode	Basic Properties to Coercive Force (Н_с_) Relationships			Average Number of Cycles to Failure (N_cycle_)
	σ_B_, MPa	σ_0.2_, MPa	HB	
As-cast state	450–520	430–500	325–337	990–1640
Post double normalizing	510–765	555–850	296–330	4460–8000
Post triple normalizing	520–690	560–795	269–310	1050–1960
Annealed	425–585	535–630	244–272	1610–2950

Note: σ_B_–ultimate tensile strength (MPa); σ_0.2_–ultimate bending strength (MPa); HB–Brinell Hardness; N_cycle_–thermal resistance was defined as a number of cycles to failure (N_cycle_). The metal was cast at 1450–1465 °C. Double normalizing at 950 ± 10 °С, 850 ± 10 °С; triple–950 ± 10 °С, 900 ± 10 °С, 850 ± 10 °С; annealing at 850 ± 10 °С. The conditioning and heating rates corresponded to the rolls’ processing parameters under production conditions within a total amount of time depending on their standard size—210 h, 140 h, and 75 h respectively.

**Table 5 materials-15-08552-t005:** Fracture surface of the test steel after the double normalizing.

Cast No.	Fracture Surface, %	
	Coarse-Crystalline	Fine-Crystalline
1	30	70
2	35	65
3	15	85
4	35	65
5	30	70
6	30	70

## Data Availability

Not applicable.

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
