# Peer review of "Improving the Performance Properties of Eutectoid Steel Products by a Complex Effect"

_materials, 2022, doi:10.3390/ma15238552_

Round 1

Reviewer 1 Report

This paper deals with the improvement in properties of a eutectoid steel subjected to different heat treatment conditions. The microstructural characteristics in different areas are also investigated. The work is interesting, but the quality of the paper should be largely improved before the paper could be considered.

Here are some suggestions/remarks (non-exhaustively listed):

-        The title of the paper is too long, and it should be reduced.

-        For the increase in thermal resistance for the V-modified steel, what is the underlying reason? It would be better if the authors can give some explanation about this in the paper.

-        Line 116: “…which is associated with an increased tendency of the metal to cracking”, this is not clear for the readers and the authors should explain a little more.

-        Figure 2 is wrongly indicated as Figure 1

-        Hc is the thermal resistance? It is better to clearly indicate this when Hc is cited in the paper for the first time.

-        In Table 3: There is a problem because “thermal resistance” and “number of cycles to failure” are together (in the same column).

-        Sigma_w and Sigma_ben are two mechanical characteristics, but what do Sigma_w and Sigma_ben represent exactly?

-        What does KC mean?

-        In Table 3: there is the presence of “number of cycles to failure”. It means that the samples have been tested under fatigue loading? If it is the case, what is the test condition? This information should be given, as the authors decided to present these results.

-        In Table 4: “tangent of the slope...” is not good. It is just the slope, since the equations represent straight lines.

-        In Table 4: What do Sigma_B and Sigma_3T mean?

-        In Table 5: What is the definition of “Fracture property (%)”? What is it exactly?

-        Line 218: After M. Goldstein, the corresponding reference should be given.

-        Line 219: “…this hardening also contributes to the maximum embrittlement of the metal”: where is the embrittlement property presented? what is the link between the hardness and the embrittlement of the metal?

Other general remarks:

While presenting the properties, it is better to interpret by taking into account (even a little) the microstructure.

Author Response

Corrections were made to the article in accordance with the reviewer's recommendations. The attached file lists the line numbers where corrections were made.

Reviewer 2 Report

The contribution is scientific. It presents a number of valuable results. The realization of the experiments was laborious, demanding, which I evaluate positively.

- In the methodology of the experiments, the used research methods, sample sizes, test equipment and the technique used in the analyzes should be defined.

- Presented results in Tab. 2 should probably be found in chapter 3. Results and Discussion.

- In the post there are 2 images marked as Figure 1. In line 114 and in line 160. I recommend checking the numbering of the images and aligning them with the text.

- What chemical analyzer was used to analyze the distribution of chemical elements?

- To fig. 3 mark what is Mn sulfide in the microstructure? so that the reader can see it immediately. Similarly, in other figs.

- To Table. 1 fill in what % of the unit is involved (wt.; atm...) The Fe content should also be filled in. (like the rest).

- How was the temperature of metals monitored in the interval 1495-1510 ºC? With what equipment and with what accuracy?

- How was the temperature of the disk 800+/-20ºC monitored?

-On which equipment were the tribological tests carried out?

- In line 257 - coagulated carbide of what?

- On what basis do you claim in the beginning of chapter 3 that the Si content was the reason for the reduction in wear? The increase of Mn, the addition of N and V probably played its role. It would be appropriate to support your statement with some study and literature.

Author Response

(The authors gave the same response as above.)

Round 2

Reviewer 1 Report

The paper is still difficult to understand. It seems that there is a logical problem between sentences and between paragraphs. The quality of writing should thus be greatly improved.
For example, the abstract and the parts 2 and 3 of the conclusions are exactly the same, which is unacceptable.

Author Response

Corrections have been made to the text, so I am attaching the entire revised manuscript. 

Reviewer 2 Report

Since the comments were accepted and the required information was added, I have no further comments on the submitted manuscript.

Author Response

(The authors gave the same response as above.)
